# Comprehensive Identification of Ginsenosides in the Roots and Rhizomes of *Panax ginseng* Based on Their Molecular Features-Oriented Precursor Ions Selection and Targeted MS/MS Analysis

**DOI:** 10.3390/molecules28030941

**Published:** 2023-01-17

**Authors:** Hong-Ping Wang, Zi-Jian Wang, Jing Du, Zhao-Zhou Lin, Chen Zhao, Run Zhang, Qiong Yin, Chun-Lan Fan, Ping Peng, Zhi-Bin Wang

**Affiliations:** 1Scientific Research Institute of Beijing Tongrentang Co., Ltd., Beijing 100011, China; 2Beijing Tongrentang Technology Development Co., Ltd., Beijing 100079, China; 3Beijing Zhongyan Tongrentang Pharmaceutical R & D Co., Ltd., Beijing 100000, China

**Keywords:** ginsenosides, LC-MS, *panax ginseng*, qualitative analysis, new strategy

## Abstract

*Panax ginseng* is widely used in Asian countries and its active constituents—ginsenosides—need to be systematically studied. However, only a small part of ginsenosides have been characterized in the roots and rhizomes of *panax ginseng* (RRPG) up to date, mainly because of a lack of the fragmentation ions of many more ginsenosides. In order to comprehensively identify ginsenosides in RRPG, molecular features of ginsenosides orienting precursor ions selection and targeted tandem mass spectrometry (MS/MS) analysis strategy were proposed in our study, in which the precursor ions were selected according to the molecular features of ginsenosides irrespective of their peak abundances, and targeted MS/MS analysis was then performed to obtain their fragmentation ions for substance characterization. Using this strategy, a total of 620 ginsenosides were successfully characterized in RRPG, including 309 protopanaxadiol-type ginsenosides, 258 protopanaxatriol-type ginsenosides and 53 oleanane-type ginsenosides. It is worth noting that, except for the known aglycones mass-to-charge ratio (*m/z*) 459, 475 and 455, twelve other aglycones, including *m/z* 509, 507, 493, 491, 489, 487, 477, 473, 461, 457, 443 and 441, were first reported in our experiment and they were probably the derivatizations of the protopanaxatriol and protopanaxadiol. Our study will not only help people to improve the cognition of ginsenosides in RRPG, but will also play a guiding and reference role for the isolation and characterization of potentially new ginsenosides from RRPG.

## 1. Introduction

Known as the king of herbs and a tonic, *panax ginseng* is widely used in Asian countries. With the development of evidence-based medicine and/or evidence-based pharmacy, analysis and identification of active constituents in *panax ginseng* are fundamental sciences needing to be addressed and systematically studied. Some modern research relevant to biological activities and pharmacological actions of *panax ginseng* have been reported, such as anti-tumor [1], improvement of immune function [2], regulation of central nervous system [3], protection effect on cerebral ischemia [4], anti-aging [5], etc. Based on a large number of published papers, the above-mentioned biological activities and pharmacological actions are all correlated with ginsenosides, a group of components that are regarded as the characteristic constituents of the roots and rhizomes of *panax ginseng* (RRPG), needing to be comprehensively analyzed.

Until now, there are mainly two approaches for studies on chemical substances of ginsenosides from RRPG. One is the chemical separation method, in which ginsenoside monomer is obtained from the extract of RRPG through separation and purification technology, and its exact structure can be concluded according to its spectral data [6]. However, some ginsenosides with minor amounts will be lost during the progress of separation, and usually ginsenosides with high amounts will be separated and characterized, leading to only a small part of ginsenosides being obtained from RRPG. For example, only eight ginsenosides were isolated from 75% ethanol extract of RRPG [7], whereas 28 ginsenosides were obtained from 70% ethanol extract of RRPG [6]. Thus far, only about 53 ginsenosides were obtained from RRPG [8]. The other approach is identifying ginsenosides based on liquid chromatography coupled with mass spectrometry (LC-MS). Especially with the help of high-resolution mass spectrometry, we can obtain the accurate molecular weights of ginsenosides as well as their molecular formulas, which is of great significance for their accurate characterization, and their structures are deduced according to the obtained tandem mass spectrometry (MS/MS) fragmentation ions. Moreover, ginsenosides with low contents or minor amounts can be characterized so long as we obtain their fragmentation information. This approach can help us to characterize much more ginsenosides in RRPG. For example, 95 ginsenosides were identified from dried fresh ginseng [9] whereas 131 ginsenosides were tentatively assigned in RRPG using ultra-high-performance liquid chromatography coupled with a diode-array detector and quadrupole/time of flight tandem mass spectrometry [10]. However, the identified ginsenosides through the above-mentioned two approaches were only a small part but not all in RRPG up to date. The main reason is that ginsenosides in RRPG own similar chemical structures and polarity leading to their poor separation in reversed-phase chromatographic column, even if in the optimized chromatographic conditions, it is difficult to achieve complete separation for many peaks and even some peaks are covered by others with high abundances. Moreover, during acquisition of their fragmentation ions, the auto-MS/MS mode is usually adopted, where precursor ions are selected according to their intensities, resulting in only compounds with high intensities having the chance to be characterized, whereas constituents with lower or much lower abundances miss being characterized because their precursor ions are not selected so their MS/MS spectra cannot be obtained. Thus, it can be seen that obtaining MS/MS spectra of much more ginsenosides during MS/MS acquisition is a crucial problem for the comprehensive identification of ginsenosides in RRPG. However, relatively few studies specially address the issues of improving acquisition of MS/MS spectra of much more ginsenosides in order to comprehensively characterize them. To solve this problem, molecular features of ginsenosides orienting precursor ions selection and targeted MS/MS analysis strategy was proposed in our study, and the precursor ions were selected according to the molecular features of ginsenosides irrespective of their peak abundances (as opposed to automatically fragmenting the most abundant signals). As a result, a total of 620 ginsenosides were successfully characterized in RRPG using this strategy.

## 2. Results

### 2.1. Data Mining

The data mining workflow mainly involved four steps (shown in Figure 1):

#### 2.1.1. Obtaining Molecular Features of Ginsenosides

The ginsenosides isolated from RRPG were used to analyze their molecular features, and in this step, the following tasks involved: (1) Consulting the literatures: the isolated ginsenosides reported in a lot of the literature were consulted and their names, structures and molecular formulas were recorded. (2) Processing the recorded data: the accurate mass of the [M – H]^−^ ion for each compound was calculated by ChemDraw software (version 14.0) based on the corresponding molecular formula, and the calculated value was saved with four decimal places. (3) Summarizing the molecular features of ginsenosides: from the recorded ginsenosides, we found that the molecular weights of ginsenosides were all above 600, indicating compounds with molecular weights below 600 probably were non-ginsenosides. The element compositions of ginsenosides were C, H, O, containing no nitrogen atom, leading to their [M – H]^−^ ions were with odd normal masses. Thus, the ions with even normal masses were probably non-ginsenosides. Furthermore, the first number after the decimal point of their [M – H]^−^ ions was between four and seven suggesting that of a compound out of this range was non-ginsenoside. The name, structure, molecular formula, molecular weight and accurate value of [M – H]^−^ ion of each isolated ginsenoside were detailed in Appendix A, ESI +.

#### 2.1.2. Extracting All Compounds in RRPG

Compounds in RRPG were extracted by Compound Discoverer software (Thermo Scientific, version 3.2.0.421) through the designed workflow mainly involved “input files→select spectra→detect compounds→group compounds”. In the step of “input files”, the full scan data of the extract solution of RRPG was selected to be processed, whereas in the step of “select spectra”, the spectra properties such as the entire retention time (0–30 min) and negative-ion mode were selected for further processing. All compounds in the raw data were extracted using the component elucidator algorithm by setting a series of parameters such as mass tolerance: 5 ppm, signal-to-noise ratio (S/N) threshold: 3, minimum peak intensity: 100,000, extracted ions: [M – H]^−^ and [M – H + HAc]^−^, minimum element composition: CHO, maximum element composition: C_90_H_190_O_90_ and so on in the “detect compounds” step. The same ingredient extracted in different addition ways was grouped by its molecular weight as well as retention time across all files in the “group compounds” step. Running this designed extracting workflow, 27,000 compounds were extracted and presented by mass-to-charge ratio (*m/z*), retention time, and peak area, and all this information was exported out as an excel file. The obtained *m/z* and retention time of compounds were absolutely essential for constructing a precursor ion list.

#### 2.1.3. Constructing Precursor Ion List of Ginsenosides

All of the extracted ions in the exported excel file were sorted by molecular weights from the smallest to the largest, and then they were filtered manually according to the molecular features of ginsenosides. The ions with molecular weights below 600, an even normal mass and the first number after the decimal point of ion not between four and seven were regarded as non-ginsenosides and filtered out. Finally, 7650 ions, which included ginsenosides, were retained to construct the precursor ion lists. However, in order to obtain high quality MS/MS spectra, the retained ions were averagely assigned in 15 precursor ion lists for segmented acquisition. Except for *m/z* and retention time (Rt), the retention time window (△Rt) was another essential parameter that was set as 0.5 min in a successful constructed list.

#### 2.1.4. Targeted MS/MS Analysis and Structure Identification

With these 15 precursor ion lists in hand, we started to analyze the extract solution of RRPG in a precursor ion-directed LC-MS/MS method and each precursor ion list was performed targeted MS/MS analysis. The ginsenosides were identified according to the obtained MS/MS spectra.

### 2.2. Ginsenosides Identification

In our experiment, a total of 620 ginsenosides (shown in Appendix A, ESI +) irrespective of their abundance were characterized in RRPG with the proposed analysis strategy including 309 protopanaxadiol-type ginsenosides, 258 protopanaxatriol-type ginsenosides as well as 53 oleanane-type ginsenosides, and all of their fragmentation ions were detailed in Appendix A, ESI +. It was worth noting that, except for the known aglycones *m/z* 459, 475 and 455, the other twelve aglycones, including aglycones *m/z* 509, 507, 493, 491, 489, 487, 477, 473, 461, 457, 443 and 441 (shown in Figure 2), were first reported in our experiment and they were probably the derivatives of the protopanaxatriol and protopanaxadiol.

Generally, the compounds in the same chemical clusters in a herbal own similar chemical structures, according to which the derivatives of the protopanaxatriol and protopanaxadiol could be deduced. The aglycone *m/z* 491 is 16 Da more than *m/z* 475 (the aglycone of protopanaxatriol) meaning it is oxidated-protopanaxatriol and its structure is probably the same as that of ginsenoside Re_5_ or ginsenjilinol (shown in Appendix A, ESI +), a known ginsenoside isolated from RRPG. However, the aglycone *m/z* 443 is 16 Da less than *m/z* 459 (the aglycone of protopanaxadiol), suggesting that it is deoxidated-protopanaxadiol and its structure is probably the same as that of vinaginsenoside R_3_ (shown in Appendix A, ESI +), a known compound isolated from the leaves of *panax ginseng*. The aglycone *m/z* 493 is 18 Da more than *m/z* 475, whereas *m/z* 457 is 18 Da less than *m/z* 475, indicating that the aglycone *m/z* 493 is double bond hydrated-protopanaxatriol with its structure probably the same as that of ginsenoside Rf_2_ (shown in Appendix A, ESI +), a known compound isolated from red *panax ginseng*, and the aglycone *m/z* 457 is dehydrated-protopanaxatriol with its structure probably the same as that of ginsenoside F_4_ (shown in Appendix A, ESI +), a known compound isolated from red *panax ginseng*, or ginsenoside Rg_6_ (shown in Appendix A, ESI +), a known compound isolated from the leaves of *panax ginseng*. In addition, the inference of the relation between the aglycone *m/z* 493 and *m/z* 475 was confirmed by ginsenosides **R75**, **R77**, **R90**, **R93** and **R95**, in whose MS/MS spectra the fragmentation ion *m/z* 475 was obtained from *m/z* 493 by losing a molecule of H_2_O. It is worth noting that, in terms of molecular weight, the aglycone *m/z* 457 is 2 Da less than *m/z* 459 probably signifying it is dehydrogenation of the protopanaxadiol. Nevertheless, we did not find any ginsenoside with the aglycone of dehydrogenated-protopanaxadiol was isolated from the roots, rhizomes or leaves of *panax ginseng*, meaning the aglycone *m/z* 457 is probably irrelevant to *m/z* 459. The aglycone *m/z* 441 is 18 Da less than *m/z* 459, indicating thtat it is dehydrated-protopanaxadiol and its structure is probably the same as that of ginsenoside Rk_1_ or ginsenoside Rg_5_ (shown in Appendix A, ESI +), a known compound isolated from red *panax ginseng*. The aglycone *m/z* 489 is 14 Da more than *m/z* 475, suggesting that it is methyl etherified-protopanaxatriol with a structure probably the same as that of ginsenoside Rh_5_ (shown in Appendix A, ESI +), a known compound isolated from the stem and leaves as well as the fruit of *panax ginseng*. The aglycone *m/z* 473 is 2 Da less than *m/z* 475, meaning that it is dehydrogenated-protopanaxatriol with a structure probably the same as that of ginsenoside Rh_7_ or ginsenoside Rh_8_ (shown in Appendix A, ESI +), a known compound isolated from the leaves of *panax ginseng*.

From the deduced aglycones mentioned above, we found that they were generally formed from protopanaxatriol or protopanaxadiol by dehydrogenation, double bond hydration, dehydration, methyl etherification, oxidation and deoxidation, based on which we deduced the aglycone *m/z* 461 was dihydrogenated-protopanaxadiol, the aglycone *m/z* 477 was dihydrogenated-protopanaxatriol, the aglycone *m/z* 509 was oxidated + hydrated-protopanaxatriol, the aglycone *m/z* 487 was methyl etherified + dehydrogenated-protopanaxatriol and the aglycone *m/z* 507 was dioxidated-protopanaxatriol.

The diagnostic ions, lost sugar moieties and substituents information played an extremely important role in ginsenosides identification. It is well known for us that there are three types of ginsenosides in RRPG, including protopanaxadiol, protopanaxatriol and oleanane, which yield the diagnostic ions at *m/z* 459, 475 and 455, respectively, in their MS/MS spectra through eliminating a series of sugar moieties. These diagnostic ions were used to deduce the types of ginsenosides. The isolated compounds (shown in Appendix A, ESI +) display the main sugar moieties in the structures of ginsenosides including glucuronic acid [Glu A (176 Da)], glucose [Glc (162 Da)], rhamnose [Rha (146 Da)], arabinose [Ara (132 Da)] as well as xylose [Xyl (132 Da)], and moreover, usually one Glu A, one or multiple Glc, one Rha, one Ara and/or one Xyl occurred on the aglycones of the ginsenosides. The non-sugar substituents in the structures mainly involved Acetyl (Ac, 42 Da), Malonyl (86 Da) and (*E*)-but-2-enoyl (68 Da), which usually do not replace other sugar moiety, but on Glc, and they are easily eliminated in the MS/MS analysis, supplying important information for ginsenosides identification.

According to the diagnostic ions, lost sugar moieties and substituents information, reference standards information, the elution order information in reversed-phase column as well as the polar size of the compound and so on, all of the ginsenosides in RRPG were successfully characterized. In addition, it is generally believed that a compound with high abundance in a full scan usually stands for a relative high amount in a herbal, which is relatively easier to be isolated than the compound with much lower abundance or amount, according to which the isolated known compounds and their isomers can be easily identified. The total ion chromatograms of the reference standards and extract solution of RRPG were shown in Appendix A.

#### 2.2.1. Characterization of Protopanaxadiol-Type Ginsenosides

##### Ginsenosides with Aglycone *m/z* 459

In our study, a total of 293 ginsenosides with aglycone *m/z* 459 were successfully identified. Except for 49 glycosylation of protopanaxadiol, which were regarded as potentially new ginsenosides, the remaining compounds were all correlated with the known ginsenosides including 110 known ginsenosides as well as their isomers, 44 acetylation, 68 malonylation, 14 (*E*)-but-2-enoylation and 8 acetylation + malonylation of known ginsenosides.

For example, eight compounds **R575**, **R585**, **R589**, **R594**, **R601**, **R604**, **R606** and **R608** eluted at 23.92 min–25.77 min showed their precursor ions at *m/z* 825.4999–825.5013 (shown in Figure 3A) with mass deviations of −0.12–1.59 ppm, suggesting that their molecular formula was C_44_H_74_O_14_. By performing their targeted MS/MS analysis, they all exhibited the same fragmentation ions and pathway. For instance, **R608**, in its MS/MS spectra, the diagnostic ion at *m/z* 459.3835 suggested it was protopanaxadiol-type ginsenoside and the mass difference between *m/z* 825.5009 and *m/z* 783.4903 was 42 Da indicating that acetyl was eliminated from the precursor ion. The fragmentation ions observed at *m/z* 621.4368 and 459.3835 indicating Glc and Glc were successively lost from *m/z* 783.4903 (shown in Figure 3B), which were the same as those of ginsenoside Rg_3_ (shown in Appendix A); therefore, **R608** and the remaining seven compounds (**R575**, **R585**, **R589**, **R594**, **R601**, **R604**, and **R606**) were tentatively assigned as acetyl-ginsenoside Rg_3_. Their extraction ion chromatograms were shown in Figure 3C, and there was a big difference in their abundances. Three compounds **R299**, **R317** and **R439** also exhibited their precursor ions at *m/z* 825.4988–825.5000 with mass deviations of −1.45–0.00 ppm and owned the same fragmentation ions; however, their diagnostic ion and fragmentation pathway were different from the above eight ginsenosides, such as **R439** (shown in Figure 3D). In its MS/MS spectra, the diagnostic ion at *m/z* 475.3794 indicating it was a protopanaxatriol-type ginsenoside and an acetyl group was eliminated due to there was a mass difference 42 Da between the precursor ion *m/z* 825.5000 and the ion *m/z* 783.4904. The ions at *m/z* 637.4363 and 475.3794 suggesting Rha and Glc were successively eliminated from the ion *m/z* 783.4904, which were the same as those of ginsenoside Rg_2_ (shown in Appendix A), therefore, **R439** and **R299**, **R317** were tentatively characterized as acetyl-ginsenoside Rg_2_. After eliminating malonyl (86 Da) from the precursor ion *m/z* 1193.5947, the remaining fragmentation ions of the fifteen compounds **R205**, **R223**, **R279**, **R292**, **R320**, **R355**, **R373**, **R412**, **R432**, **R440**, **R448**, **R479**, **R485**, **R529** and **R546** were observed at *m/z* 1107.5950, 945.5453, 783.4878, 621.4369 as well as 459.3831, which were the same as those of ginsenoside Rb_1_. **R279** was deduced as a known constituent malonyl-ginsenoside Rb_1_ whereas the others were assigned as its isomers due to the relative abundance of **R279** was much higher than the others. After eliminating (*E*)-but-2-enoyl (68 Da) from the precursor ion *m/z* 1013.5687, the remaining fragmentation ions of the four compounds **R528**, **R558**, **R567** and **R586** were detected at *m/z* 945.5403, 783.4888, 621.4343 as well as 459.3857, which were the same as those of ginsenoside Rd (shown in Appendix A), therefore, the four compounds were tentatively characterized as (*E*)-but-2-enoyl ginsenoside Rd. Thus it can be seen, the acetylation, malonylation, and (*E*)-but-2-enoylation of the known ginsenosides will first eliminate acetyl, malonyl and (*E*)-but-2-enoyl in their MS/MS spectra and the remaining fragmentation ions are the same as those of their corresponding ginsenosides, which can be used to deduce other acetyl-, malonyl-, and (*E*)-but-2-enoyl-ginsenosides with aglycone *m/z* 459.

Besides, some potential new ginsenosides with aglycone *m/z* 459 were also characterized. For instance, **R219** (shown in Figure 3E), its precursor ion was extracted at *m/z* 1269.6471 with a mass deviation of –0.63 ppm indicating its molecular formula was C_60_H_102_O_28_. The diagnostic ion was at *m/z* 459.3835, indicating that it was a protopanaxatriol-type ginsenoside. The fragmentation ions at *m/z* 1107.5922, 945.5416, 783.4902, 621.4380 and 459.3835 suggested Glc, Glc, Glc, Glc and Glc were successively eliminated from the precursor ion, respectively. However, the current known compounds were not well matched with **R219**, therefore, **R219** was tentatively characterized as a potential new ginsenoside protopanaxadiol + 5Glc. For **R288**, the mass difference between the precursor ion *m/z* 1355.6470 and its fragmentation ion *m/z* 1269.5906 was 86 Da indicated that malonyl was eliminated from the precursor ion. The remaining fragmentation ions (shown in Figure 3F) were the same as those of **R219**, thus **R288** was probably malonylation of **R219**. However, according to the known malonylation of ginsenosides, malonyl always occurs on Glc, thus **R288** was tentatively characterized as protopanaxadiol + 4Glc + malonyl Glc. Similarly, other 47 potential new ginsenosides with aglycone *m/z* 459 were assigned.

##### Ginsenosides with Aglycone *m/z* 461

The aglycones of **R120**, **R138** and **R155** were *m/z* 461, which was deemed as dihydrogenated-protopanaxadiol due to it being 2 Da more than *m/z* 459. For example, **R138** (shown in Figure 4A), its precursor ion, was extracted at *m/z* 1121.5742 with a mass deviation of –0.18 ppm, indicating that its molecular was C_54_H_90_O_24_. The mass difference between the precursor ion and its fragmentation ion *m/z* 1079.5619 was 42 Da suggested that acetyl was lost in MS/MS spectra. The ions observed at *m/z* 947.5371, 785.4726, 623.4160 and 461.3647 were formed by successive losses of Xyl/Ara, Glc, Glc and Glc from the ion *m/z* 1079.5619. Generally, the acetylation always occurs on Glc from the known acetyl-ginsenosides, thus **R138** was deduced as dihydrogenated-protopanaxadiol + Xyl/Ara + 2Glc + acetyl Glc.

##### Ginsenosides with Aglycone *m/z* 443

Two compounds, **R381** and **R169,** were elucidated as ginsenosides with aglycone *m/z* 443. For example, the precursor ion of **R169** was extracted at *m/z* 1253.6544 and generated the aglycone at *m/z* 443.3906 by successive losses of Glc, Glc, Glc, Glc, Glc in its MS/MS spectra (shown in Figure 4B). Thus, **R169** was deduced as an unknown ginsenoside deoxidated-protopanaxadiol + 5Glc. The aglycone of **R381**, *m/z* 443.3901, was formed by successive losses of Glc, Glc and Glc from the precursor ion *m/z* 929.5488. **R381** was tentatively deduced as a known compound vinaginsenoside R_3_.

##### Ginsenosides with Aglycone *m/z* 441

A total of five compounds, including **R226**, **R443**, **R446**, **R616** and **R618,** were characterized as ginsenoisdes with aglycone *m/z* 441. Three compounds, **R226**, **R443**, and **R446**, owned not only the same precursor ions but also the same fragmentation ions. Taking **R446** as an example, **R446** was extracted at *m/z* 807.4531 with mass deviations of 0.00 ppm indicating its molecular formula was C_43_H_68_O_14_. In its MS/MS spectra (shown in Figure 4C), the mass difference between the fragmentation ion *m/z* 765.4421 and the precursor ion *m/z* 807.4531 was 42 Da indicating acetyl was eliminated. The ion at *m/z* 603.3807 and 441.3345 were formed by successive losses of Glc and Glc from the ion *m/z* 765.4421, respectively, indicating there were 2Glc in its structure. Due to acetyl always occurs on Glc, **R446** were elucidated as dehydrated-protopanaxadiol + acetyl Glc + Glc. The precursor ions of **R616** (eluted at 26.72 min) and **R618** (eluted at 26.90 min) were separately extracted at *m/z* 765.4795 and 765.4798. By performing their targeted MS/MS analysis, we found that they owned the same fragmentation ion *m/z* 603.4263 and aglycone *m/z* 441.3346, which were formed by successively eliminating Glc and Glc from the precursor ion, respectively. **R616** and **R618** were tentatively assigned as ginsenoside Rk_1_ and ginsenoside Rg_5_, respectively. As ginsenoside Rk_1_ was eluted before ginsenoside Rg_5_ on a reverse-phase column, **R616** was deduced as ginsenoside Rk_1_, whereas **R618** was characterized as ginsenoside Rg_5_.

#### 2.2.2. Characterization of Protopanaxatriol-Type Ginsenosides

##### Ginsenosides with Aglycone *m/z* 475

In our experiment, a total of 145 ginsenosides were identified with aglycone *m/z* 475, including 33 glycosylation of protopanaxatriol, 69 known ginsenosides, as well as their isomers, 20 acetylation, 19 malonylation, 2 (*E*)-but-2-enoylation and 2 acetylation + malonylation of known ginsenosides. Except for that, six ginsenosides (**R66** and its isomers **R72**, **R103**, **R112**, **R124** and **R152**) belong to protopanaxadiol-type also exhibited their aglycones at *m/z* 475.

For instance, the precursor ions of eight compounds **R106**, **R114**, **R118**, **R127**, **R141**, **R151**, **R171** and **R175** were extracted at *m/z* 987.5519–987.5536 (shown in Figure 5A) with mass deviations of −1.01–0.71 ppm indicating their molecular formula was C_50_H_84_O_19_. In their MS/MS spectra, they all showed the same fragmentation ions, taking **R127** as an example. We deduced an acetyl was eliminated from the precursor ion due to the ion *m/z* 945.5443 was observed, and then Glc, Rha and Glc were successively eliminated because a series of ions *m/z* 783.4902, 637.4315 and 475.3795 were obtained (shown in Figure 5B). By observing carefully, after loss of acetyl, the fragmentation ions and pathway were the same as those of ginsenoside Re (shown in Appendix A), thus, **R127** and the other seven compounds **R106**, **R114**, **R118**, **R141**, **R151**, **R171**, **R175** were deduced as acetyl-ginsenoside Re. Their extraction ion chromatograms were shown in Figure 5C and we found there was a big difference in their abundances. It is worth noting that, although the twelve compounds **R364**, **R382**, **R389**, **R429**, **R460**, **R476**, **R499**, **R506**, **R535**, **R545**, **R613** and **R617** also exhibited the precursor ion at *m/z* 987.5524–987.5538 (shown in Figure 5A), their fragmentation ions were different from those of acetyl-ginsenoside Re. And the twelve compounds owned the same fragmentation ions, such as **R535** (shown in Figure 5D). After eliminating the acetyl group from the precursor ion *m/z* 987.5537, the fragmentation ion *m/z* 945.5444 was obtained, and it subsequently generated the ions *m/z* 783.4902, 621.4377 and 459.3849, which were the same as those of ginsenoside Rd (shown in Appendix A). Based on the above information, **R535** and the other eleven compounds were characterized as acetyl-ginsenoside Rd. Thus, it can be seen that the diagnostic ions played an extremely important role in the characterization of the ginsenosides in RRPG.

In the same way, other known ginsenosides with aglycone *m/z* 475 as well as their isomers, acetylations, malonylations, (*E*)-but-2-enoylations were tentatively assigned and the fragmentation pathways were used to deduce the 33 unknown ginsenosides with aglycone *m/z* 475, namely glycosylation of protopanaxatriol, such as **R202**. The mass difference between the precursor ion *m/z* 1179.5806 and its fragmentation ion *m/z* 1093.5795 was 86 Da, suggested that malonyl group was eliminated from the precursor ion in its MS/MS spectra. The ions at *m/z* 961.5365, 799.4838, 637.4296, 475.3789 suggested Ara/Xyl, Glc, Glc and Glc were successively eliminated from the ion *m/z* 1093.5795, respectively (shown in Figure 5E). The diagnostic ion *m/z* 475.3789 indicated that **R202** was protopanaxatriol-type ginsenosides.

However, current known ginsenosides were not well matched with it, thus **R202** was deemed as a new ginsenoside protopanaxatriol + Ara/Xyl + 2Glc + Malonyl Glc. The precursor ion of **R224** was extracted at *m/z* 859.4685 with a mass deviation of –0.70 ppm and in its MS/MS spectra, the mass difference between the ion *m/z* 813.4393 and the precursor ion was 46 Da suggesting *m/z* 859.4685 was [M – H + HAc]^–^ ion whereas *m/z* 813.4393 was [M – H]^–^ ion. The fragmentation ions *m/z* 637.4297, 475.3783 suggested Glu A and Glc were successively eliminated from [M – H]^–^ ion, respectively (shown in Figure 5F). **R224** was characterized as a new ginsenoside protopanaxatriol + Glu A + Glc. In the same way, the other 31 potential new ginsenosides with aglycone *m/z* 475 were tentatively characterized.

##### Ginsenosides with Aglycone *m/z* 509

The aglycones of **R1**, **R2** and **R16** were *m/z* 509, which was 34 Da more than *m/z* 475, suggesting it was oxidated + hydrated-protopanaxatriol. **R1** showed its [M–H]^–^ ion at *m/z* 979.5486 (mass deviation of 0.82 ppm) while **R2** exhibited its [M – H + HAc]^–^ ion at *m/z* 1025.5530 (mass deviation of –0.20 ppm), which indicated that the molecular formula of **R1** and **R2** was C_48_H_84_O_20_. In their MS/MS spectra, they owned the same fragmentation ions, such as **R1** (shown in Figure 6A). The ions at *m/z* 817.4918, 671.4409 and 509.3840 were formed by successive losses of sugar moieties Glc, Rha and Glc from the precursor ion, respectively. In addition, a series of dehydrated ions were observed at *m/z* 799.4863, 653.4276, 635.4174 and 491.3755. Thus, **R1** and **R2** were deduced as oxidated + hydrated-protopanaxatriol + 2Glc + Rha. After eliminating Glc, the remaining fragmentation ions of **R1** or **R2** were the same as those of **R16**, which meant that the structure of **R16** was one Glc less than that of **R1** or **R2**. Based on the above information, **R16** was deemed as oxidated + hydrated protopanaxatriol + Glc + Rha.

##### Ginsenosides with Aglycone *m/z* 507

The aglycone of **R12** was observed at *m/z* 507.3710, which was deemed as dioxidated-protopanaxatriol due to it being 32 Da more than *m/z* 475. In its MS/MS spectra, the mass difference between the precursor ion *m/z* 877.4809 and the fragmentation ion *m/z* 831.4691 was 46 Da implying *m/z* 877.4809 was [M – H + HAc]^–^ ion whereas *m/z* 831.4691 was [M – H]^–^ ion. There were 2Glc in its structure because the aglycone was formed by losses of 2Glc from [M – H]^–^ ion (shown in Figure 6B), thus **R12** was elucidated as dioxidated-protopanaxatriol + 2Glc.

##### Ginsenosides with Aglycone *m/z* 493

A total of 20 compounds were assigned as ginsenosides with aglycone *m/z* 493. For example, the [M – H]^–^ ions of **R90** was extracted at *m/z* 1021.5587. In its MS/MS spectra, the fragmentation ions at *m/z* 979.5484, 817.4941, 655.4421, and 493.3891 were formed by successive losses of Ac, Glc, Glc, and Glc from the precursor ion, respectively (shown in Figure 6C). Due to the aglycone of **R90** being double bond hydrated-protopanaxatriol, there were much more hydroxyl groups in its structure leading to much more dehydrated ions were obtained. Finally, **R90** was characterized as double bond hydrated-protopanaxatriol + 2Glc + acetyl Glc. The precursor ions ([M – H + HAc]^–^) of **R36** and **R47** were extracted at *m/z* 847.5063 and 847.5060, respectively, with mass deviations of no more than 0.94 ppm. By performing their targeted MS/MS analysis, they all generated the same fragmentation ions at *m/z* 801.5004, 655.4425 and 493.3891 by successive losses of HAc, Rha and Glc from the precursor ion, respectively. **R36** was deduced as a known compound ginsenoside Rf_2_ whereas **R47** was elucidated as its isomer due to the relative abundance of **R36** being much higher than that of **R47**. In the same way, other ginsenosides with aglycone *m/z* 493 were elucidated.

##### Ginsenosides with Aglycone *m/z* 491

In our study, a total of 36 compounds were identified as ginsenosides with aglycone *m/z* 491. Four compounds, **R3**, **R8**, **R9** and **R11**, showed their [M – H + HAc]^–^ precursor ions at *m/z* 861.4847–861.4856 with mass deviations of −0.12–0.93 ppm, whereas the fourteen compounds **R15**, **R17**, **R18**, **R20**, **R24**, **R41**, **R57**, **R70**, **R99**, **R119**, **R139**, **R153**, **R170** and **R193** exhibited their [M – H]^–^ precursor ions at *m/z* 815.4790–815.4804 with mass deviations of −0.37–1.35 ppm. In their MS/MS spectra, all the eighteen compounds shared the same fragmentation ions at *m/z* 653.4243 and 491.3737 formed by eliminating Glc and Glc from their [M – H]^–^ ion, respectively. However, **R20** was assigned as ginsenoside Re_5_ whereas **R70** was deemed as ginsenjilinol by comparison with the reference standards and subsequently the other sixteen compounds were characterized as their isomers. From Appendix A, ESI +, we found that the aglycones of ginsenoside Re_5_ and ginsenjilinol were with different structures due to their cis-trans of double bond in C24 and C25 being different, but all were oxidated-protopanaxatriol. Thus, we consider that other identified ginsenosides with aglycone *m/z* 491 probably own the same aglycone structure as either ginsenoside Re_5_ or ginsenjilinol. For instance, **R85** (shown in Figure 6D) was deduced as oxidated-protopanaxatriol + acetyl Glc + Rha because its aglycone was obtained at *m/z* 491.3756 and was formed by successive losses of Ac, Rha and Glc from the precursor ion *m/z* 841.4953. Similarly, the other 17 compounds with aglycone *m/z* 491were tentatively characterized.

##### Ginsenosides with Aglycone *m/z* 489

Six compounds, **R10**, **R13**, **R31**, **R51**, **R62** and **R126,** were deduced as ginsenosides with aglycone *m/z* 489. For example, the [M – H + HAc]^–^ precursor ion of **R126** was extracted at *m/z* 8159.4701. In its MS/MS spectra (shown in Figure 6E), the fragmentation ions at *m/z* 651.4153 and 489.3582 were formed by successive losses of Glc and Glc from [M – H]^–^ ion, respectively. Thus, **R126** was elucidated as methyl etherified-protopanaxatriol + 2Glc.

##### Ginsenosides with Aglycone *m/z* 487

The aglycone of **R62** was at *m/z* 487, which was deemed as methyl etherified + dehydrogenated-protopanaxatriol because it was 12 Da more than *m/z* 475. As shown in Figure 6F, the aglycone *m/z* 487.3440 was formed by successive losses of Glc and Glu A from the precursor ion *m/z* 987.4805. Thus, **R62** was concluded as methyl etherified + dehydrogenated-protopanaxatriol + 2Glc + Glu A.

##### Ginsenosides with Aglycone *m/z* 477

The aglycones of the three compounds, **R91**, **R148** and **R156**, were *m/z* 477, which was deemed as dihydrogenated-protopanaxatriol due to it being 2 Da more than *m/z* 475. For **R91** (shown in Figure 6G), its [M – H + HAc]^−^ ion was extracted at *m/z* 1009.5579 which was confirmed by the [M – H]^–^ ion observed at *m/z* 963.5515 in its MS/MS spectra because their mass difference was 46 Da. The aglycone at *m/z* 477.3950 was formed by successive losses of Glc, Glc and Glc from [M – H]^–^ ion, suggesting that there was 3Glc in its structure. Thus, **R91** was deduced as dihydrogenated-protopanaxatriol + 3Glc. In the same way, other ginsenosides with aglycone *m/z* 477 were characterized.

##### Ginsenosides with Aglycone *m/z* 473

A total of 20 compounds, including **R39**, **R67**, **R80**, **R102**, **R129**, **R133**, **R159**, **R161**, **R179**, **R183**, **R186**, **R187**, **R194**, **R199**, **R241**, **R246**, **R333**, **R480**, **R452** and **R532**, were tentatively deduced as ginsenosides with aglycone *m/z* 473. **R452** and **R532** separately eluted at 21.39 min and 22.79 min were deduced as two known compounds ginsenoside Rh_7_ and ginsenoside Rh_8_ due to their aglycones at *m/z* 473.3669 being formed by a loss of Glc from their [M – H]^–^ ions. However, according to their structures, ginsenoside Rh_7_ is with higher polarity than ginsenoside Rh_8_ leading to it is eluted firstly on a reverse-phase column. Thus, **R452** was assigned as ginsenoside Rh_7_, whereas **R532** was deduced as ginsenoside Rh_8_. Although the aglycones of ginsenoside Rh_7_ and ginsenoside Rh_8_ were with different structures due to their different dehydrogenated positions, actually they were dehydrogenated-protopanaxatriol. **R129** was deduced as dehydrogenated-protopanaxatriol + 3Glc because its aglycone *m/z* 473.3635 was formed by eliminating 3Glc from the precursor ion *m/z* 959.5223 (shown in Figure 6H). Similarly, other ginsenosides with aglycone *m/z* 473 were tentatively characterized.

##### Ginsenosides with Aglycone *m/z* 457

In our experiment, 23 compounds, including **R160**, **R231**, **R238**, **R262**, **R270**, **R280**, **R298**, **R340**, **R360**, **R363**, **R380**, **R388**, **R392**, **R418**, **R434**, **R450**, **R469**, **R470**, **R556**, **R566**, **R591**, **R602** and **R610,** were deduced as ginsenosides with aglycone *m/z* 457. **R556** and **R566** exhibited the same aglycone at *m/z* 457.3707 which was formed by successive losses of Rha and Glc from the precursor ion *m/z* 765.4795. **R556** was assigned as ginsenoside Rg_6_ whereas **R566** was deemed as ginsenoside F_4_, and they were confirmed by comparison with the reference standards. Their aglycones were with different structures, but all were dehydrated-protopanaxatriol. **R160** was characterized as dehydrated-protopanaxatriol + 4Glc due to the alycone *m/z* 457.3698 being formed by losses of 4Glc from the precursor ion *m/z* 1105.5779 (shown in Figure 6I). In the same way, other ginsenosides with aglycone *m/z* 457 were tentatively assigned.

#### 2.2.3. Characterization of Oleanane-Type of Ginsenosides

A total of 53 oleanane-type of ginsenosides with aglycone *m/z* 455 were identified in this study, including 33 known ginsenosides, as well as their isomers, seven acetylations, four malonylations, four (*E*)-but-2-enoylations, one acetylation + malonylation and four glycosylation of oleanolic aglycone.

For example, the precursor ions of the eight compounds, **R240**, **R253**, **R265**, **R301**, **R315**, **R385**, **R438** and **R504**, were extracted at *m/z* 955.4899–955.4916 (shown in Figure 7A) with mass deviations of −0.42–1.36 ppm, indicating that their molecular formula was C_48_H_76_O_19_. By performing their targeted MS/MS analysis, they all exhibited the same fragmentation ions, such as **R438** (shown in Figure 7B), at *m/z* 793.4358, 731.4333 and 569.3846 formed by successive losses of Glc, [CO_2_ + H_2_O] and Glc from the precursor ion *m/z* 955.4912, respectively. The aglycone at *m/z* 455.3546 was formed by loss of [2Glc + Glu A] from *m/z* 955.4912. **R301** was assigned as ginsenoside Ro and confirmed by the reference standard, **R438** and the other six compounds **R240**, **R253**, **R265**, **R315**, **R385** and **R504** were tentatively deemed as ginsenoside Ro isomers. Their extraction ion chromatograms were shown in Figure 7C, from which we found the abundance of ginsenosides Ro was much higher than those of the other seven ginsenosides. Except for the ginsenosides identified correlating with the known compounds, four unknown ginsensides, which were deemed as glycosylation of oleanolic aglycone, were also characterized. For instance, **R221** was deduced as oleanolic aglycone + 2Glc + Xyl/Ara + Glu A due to the aglycone *m/z* 455.3531 was formed by a loss of [2Glc + Xyl/Ara + Glu A] from the precursor ion *m/z* 1087.5337 (shown in Figure 7D) whereas **R346** was deemed as oleanolic aglycone + Glc + Rha + Glu A because its aglycone *m/z* 455.3532 was formed by loss of [Glc + Rha + Glu A] from the precursor ion *m/z* 939.4953 (shown in Figure 7E).

## 3. Materials and Methods

### 3.1. RRPG Sample, Reference Standards and Reagents

RRPG was supplied by the Scientific Research Institute of Beijing Tongrentang Co., Ltd. Total 35 reference standards, including ginsenoside Re_5_ (**R20**), Re_3_ (**R28**), Re_4_ (**R34**), Re_1_ (**R46**), Re_2_ (**R54**), Rg_1_ (**R73**), Re (**R74**), Rf (**R200**), Ra_2_ (**R248**), Ra_3_ (**R257**), Rb_1_ (**R259**), Ra_1_ (**R289**), Ro (**R301**), Rb_2_ (**R324**), Rb_3_ (**R335**), Rs_2_ (**R401**), Rd (**R406**), Ro methyl ester (**R493**), Rg_6_ (**R556**), F_4_ (**R566**), 20-gluco-ginsenoside Rf (**R40**), notoginsenoside R_1_ (**R50**), R_4_ (**R201**), R_2_ (**R216**), quinquenoside R_1_ (**R358**), ginsenjilinol (**R70**), 20(*S*)-ginsenoside Rg_2_ (**R261**), 20(*S*)-ginsenoside Rh_1_ (**R263**), 20(*R*)-ginsenoside Rg_2_ (**R281**), ginsenoside Rc (**R285**), 20(*R*)-ginsenoside Rh_1_ (**R286**), 20(*S*)-ginsenoside Rg_3_ (**R593**), 20(*R*)-ginsenoside Rg_3_ (**R599**), 20(*S*)-ginsenoside-Rh_2_ (**R619**) and 20(*R*)-ginsenoside-Rh_2_ (**R620**), were either purchased from the National Institute for the Control of Pharmaceutical and Biological Products (Beijing, China) as well as Shanghai Yuanye Bio-Technology Co., Ltd. (Shanghai, China) or gifts from the State Key Laboratory of Natural and Biomimetic Drugs, Department of Natural Medicines, School of Pharmaceutical Sciences, Peking University. The purity of all the reference standards were >98%. LC-MS-grade acetonitrile as well as methanol was obtained from Merck (Darmstadt, Germany), LC-MS grade formic acid was purchased from Fisher-Scientific (Fair Lawn, NJ, USA), and distilled water was obtained from Watsons.

### 3.2. Sample and Standard Solution Preparations

RRPG was pulverized into powder (the dimension of the obtained powder just like flour). The powder of RRPG (1.0 g) was extracted ultrasonically for 30 min with 10 mL 70% methanol at 25 °C. The extracted solution was filtered through a 0.22 μm nylon filter membrane before analysis. The 35 reference standards were used only for confirmation of the accuracy of the characterization rather than quantifications. Each of the reference standards (1.0 mg) was separately dissolved in 1 mL 70% methanol to obtain the stock solution (1.0 mg/mL) of each reference standard. 10–20 μL of each stock solution was taken to mix and the mixture stock solution was obtained. Then, they were stored at 4 °C until analysis.

### 3.3. Chromatographic and Mass Spectrometric Conditions

The separation of the multiple components were performed using a Vanquish™ Flex UHPLC system (Thermo Scientific, Waltham, MA, USA), equipped with a binary pump and a thermostatted column compartment. Desirable chromatographic separation of the extract of RRPG was obtained on a Waters ACQUITY UPLC^®^ BEH C_18_ column (2.1 × 100 mm, 1.7 μm) coupled with a ACQUITY UPLC^®^ BEH C_18_ VanGuard^TM^ Pre-Column (2.1 × 5 mm, 1.7 μm) by using of the mobile phase A (0.1% formic acid/water, *v/v*) and mobile phase B (acetonitrile) by the following gradient elution program: 0–7 min, 2%–20% B; 7–10 min, 20%–25% B; 10–20 min, 25%–40% B; 20–25 min, 40%–65% B; 25–30min, 65%B–95% B. The flow rate was 0.3 mL/min and the temperature was set at 35 °C. The injection volume was 2 μL.

High-accuracy mass spectrometric data were recorded on Orbitrap Exploris 240 mass spectrometer (Thermo Scientific, USA) with Heated ESI source. The instrument was operated in negative-ion mode. The optimized MS parameters were described as follows: ion spray voltage: 2500 V, sheath gas: 5.08 L/min, aux gas: 9.37 L/min, ion transfer tube temperature: 320 °C, vaporizer temperature: 350 °C, scan range (*m/z*): 150–2000 and collision-energy voltage: 35 V. The full scan was operated at a mass resolution of 60,000 whereas the MS^2^ scan was operated at a mass resolution of 15,000. Internal calibration source Thermo Scientific EASY-IC^TM^ was adopted to calibrate the entire mass range.

### 3.4. Data Acquisition and Data Processing

Data acquisition was performed on Thermo Xcalibur software (version 4.5) whereas data processing and data mining were performed on Free Style^TM^ 1.8 SP1 software and Compound Discoverer^TM^ software (Thermo Scientific^TM^, version 3.2.0.421). Negative-ion mode was adopted to perform data acquisition due to ginsenosides exhibited high [M – H]^–^ and/or [M – H + HAc]^–^ ion responses in a full scan in this mode. The extracted solution of RRPG was first analyzed in a full scan mode so as to minimize the losses of signals and then targeted MS/MS analysis was followed after constructing the precursor ion lists.

## 4. Conclusions

While there are many ginsenosides in RRPG, the ginsenosides currently isolated from it are just in part, which leads to the limitation of the cognition of ginsenosides for people. In our study, a strategy based on molecular features of ginsenosids orienting precursor ions selection and targeted MS/MS analysis was proposed to comprehensively identify ginsenosides in RRPG. The identified 620 ginsenosides confirmed the current cognition that ginsenosides in RRPG mainly existed in the forms of the known three types of protopanaxadiol, protopanaxatriol and oleanane. Furthermore, except for the known aglycones *m/z* 475, 459 and 455, our study proved that there were many more types of aglycones existing in RRPG, such as *m/z* 461, 443 as well as 441 for protopanaxadiol-type and *m/z* 509, 507, 493, 491, 489, 487, 477, 473 as well as 457 for protopanaxatriol-type, and the aglycones of protopanaxatriol-type were much more various than those of protopanaxadiol-type. Nevertheless, the oleanane-type ginsenosides were only with the aglycone *m/z* 455. Our study will help people to improve the cognition and understanding of ginsenosides in RRPG.

It is worth noting that malonyl-ginsenosides are acidic saponins with strong polarity and are extremely easily hydrolyzed in the conditions of acid, alkali, hot water as well as hot methanol or ethanol to form corresponding ginsenosides by the loss of malonyl [11]. Therefore, in order to retain the prototype of malonyl-ginsenosides, the powders of RRPG was extracted with 70% methanol at 25 °C so as to try to avoid the destabilized factor and finally much more malonyl-ginsenosides were detected, such as 108 malonyl-ginsenosides as well as 24 acetylation + malonylation ginsenosides were successfully identified.

In addition, the unknown ginsenosides identified as the isomers, acetylations, malonylations, (*E*)-but-2-enoylations, acetylation + malonylation of the known ginsenosides as well as glycosylation of protopanaxadiol, glycosylation of protopanaxatriol, glycosylation of oleanolic aglycone are probably potential new compounds. For example, ginsenjilinol (**R70**) with aglycone *m/z* 491 was previously isolated as a new ginsenoside from RRPG by our research group and owned some certain anti-inflammatory activity [12]. Thus, our study will play a guiding and reference role for the isolation and characterization of these potential new ginsenosides in RRPG. Besides, *panax ginseng*, which is a traditional Chinese medicine with clinical value and a tonic with nourishing effect, is often combined with other medicines forming many prescriptions to cure multiple diseases [13]. However, *panax ginseng* is mainly used in the form of powers and it is directly mixed with the extracts of other medicines in these prescriptions, which leads to ginsenosides in the prescriptions just come from the involved single herb *panax ginseng*. Therefore, our study will also provide a reference for ginsenosides identification for traditional Chinese formulas containing *panax ginseng*.

## Figures and Tables

**Figure 1 molecules-28-00941-f001:**
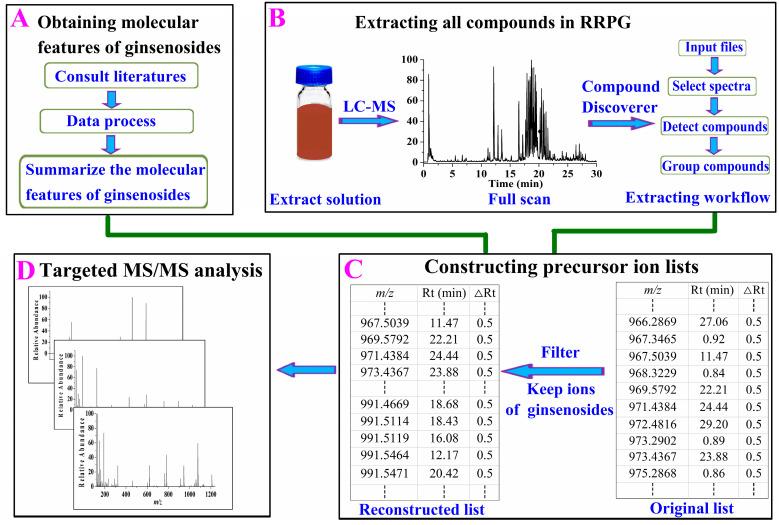
The data mining strategy proposed in our study. (**A**) Obtaining molecular features of ginsenosides; (**B**) extracting all compounds in the roots and rhizomes of *panax ginseng* (RRPG); (**C**) constructing precursor ion list; (**D**) targeted tandem mass spectrometry (MS/MS) analysis; LC-MS: liquid chromatography coupled with mass spectrometry, *m/z*: mass-to-charge ratio, Rt: retention time, △Rt: the retention time window.

**Figure 2 molecules-28-00941-f002:**
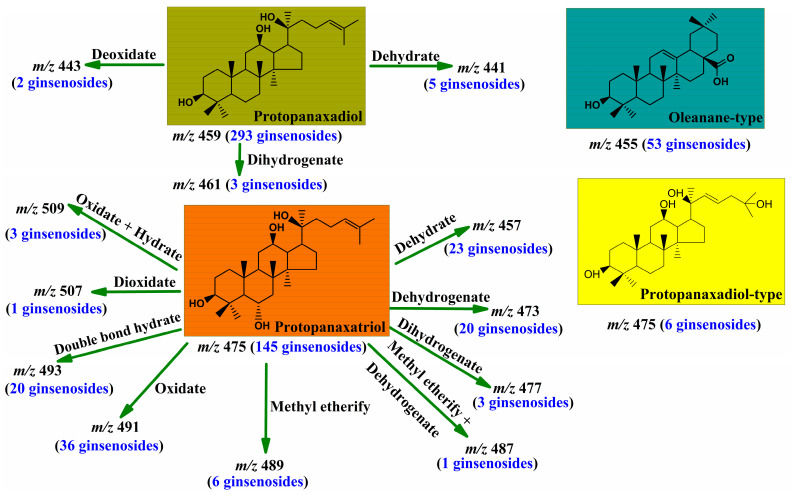
The identified 620 ginsenosides had different aglycones.

**Figure 3 molecules-28-00941-f003:**
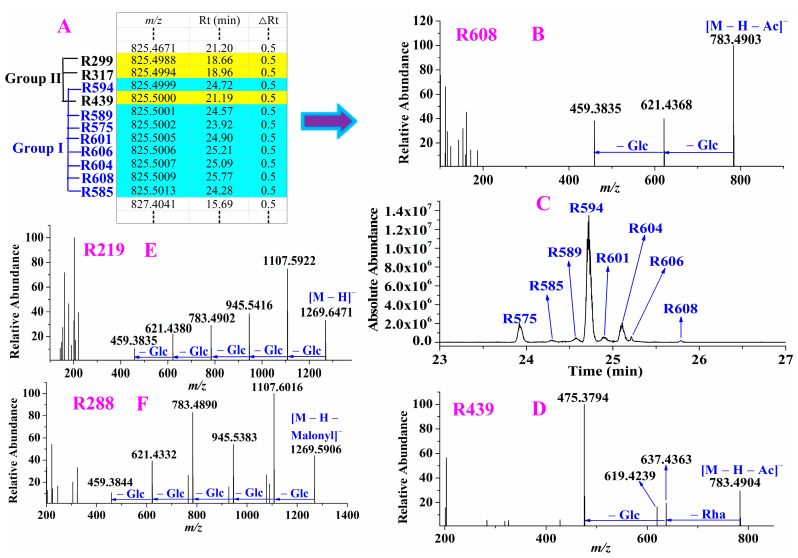
An example of the characterization of ginsenosides with aglycone *m/z* 459. The precursor ions of eleven compounds were reconstructed in list (**A**); performing targeted MS/MS analysis for compounds in Group I such as **R608** (**B**) and in Group II such as **R439** (**D**); extraction ion chromatograms of compounds in Group I (**C**); the MS/MS spectrum of potential new ginsenosides **R219** (**E**) and **R288** (**F**); Glc: glucose, Rha: rhamnose, Ac: acetyl.

**Figure 4 molecules-28-00941-f004:**
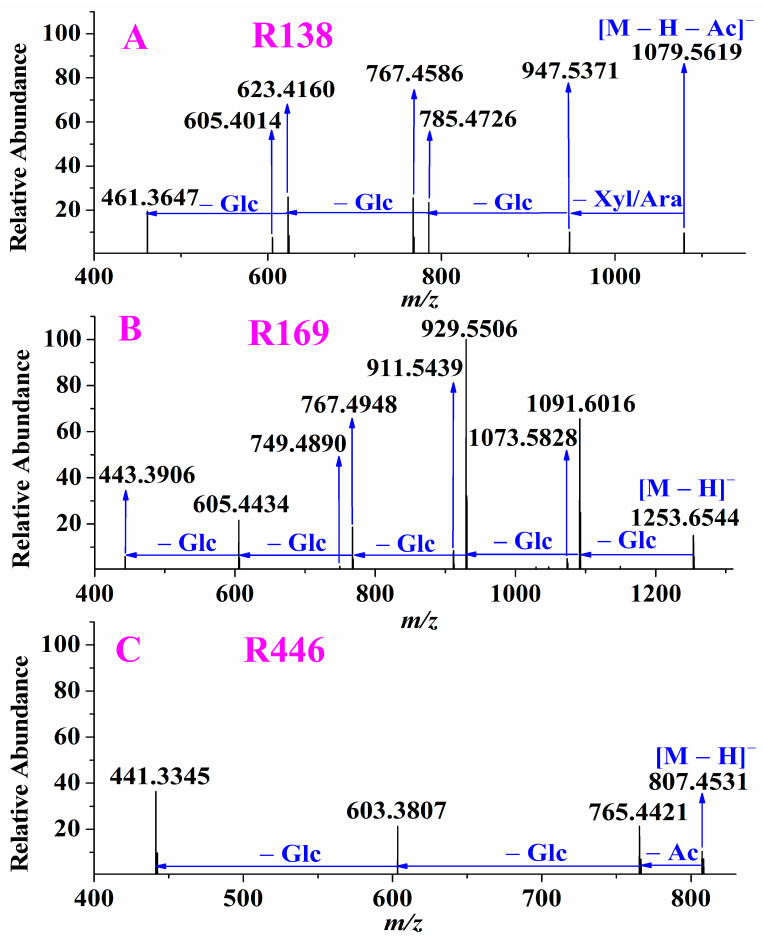
Examples of characterization of ginsenosides with different aglycones: *m/z* 461, such as **R138** (**A**); *m/z* 443, such as **R169** (**B**); *m/z* 441, such as **R446** (**C**); Xyl/Ara: xylose/arabinose.

**Figure 5 molecules-28-00941-f005:**
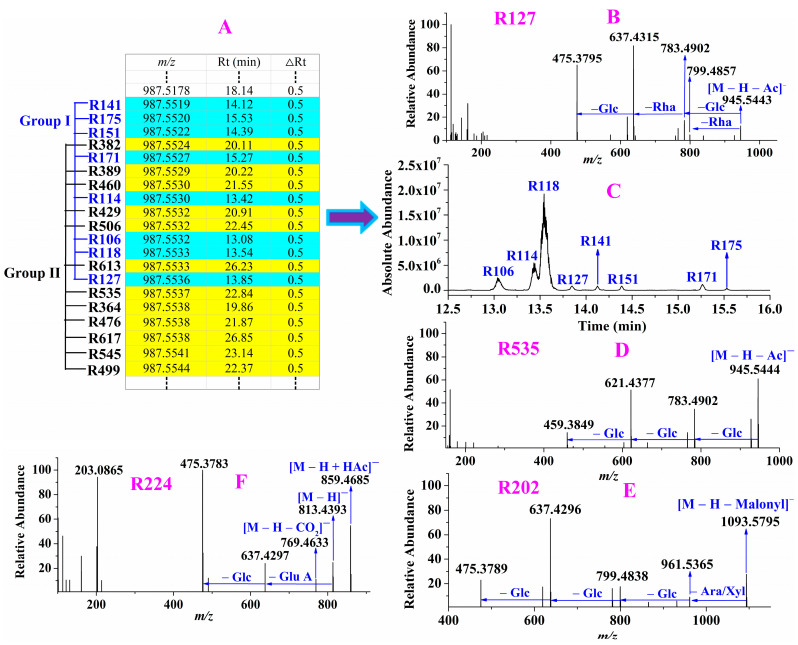
An example of characterization of ginsenosides with aglycone *m/z* 475. The precursor ions of twenty compounds were reconstructed in list (**A**); performing targeted MS/MS analysis for compounds in Group I such as **R127** (**B**) and in Group II such as **R535** (**D**); extraction ion chromatograms of compounds in Group I (**C**); The MS/MS spectrum of potential new ginsenosides **R202** (**E**) and **R224** (**F**).

**Figure 6 molecules-28-00941-f006:**
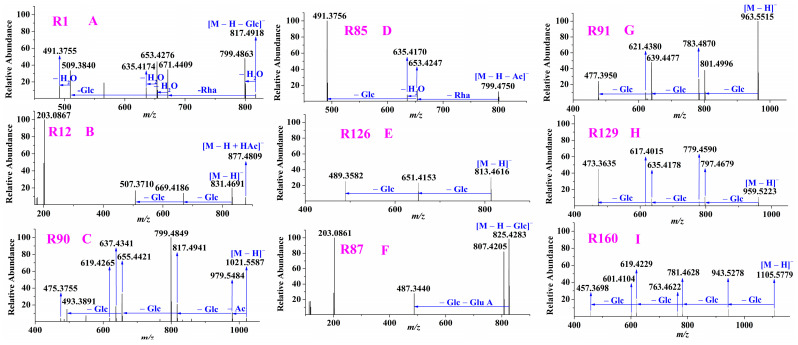
Examples of characterization of ginsenosides with aglycones *m/z* 509 (**A**); *m/z* 507 (**B**); *m/z* 493 (**C**); *m/z* 491 (**D**); *m/z* 489 (**E**); *m/z* 487 (**F**); *m/z* 477 (**G**); *m/z* 473 (**H**); *m/z* 457 (**I**).

**Figure 7 molecules-28-00941-f007:**
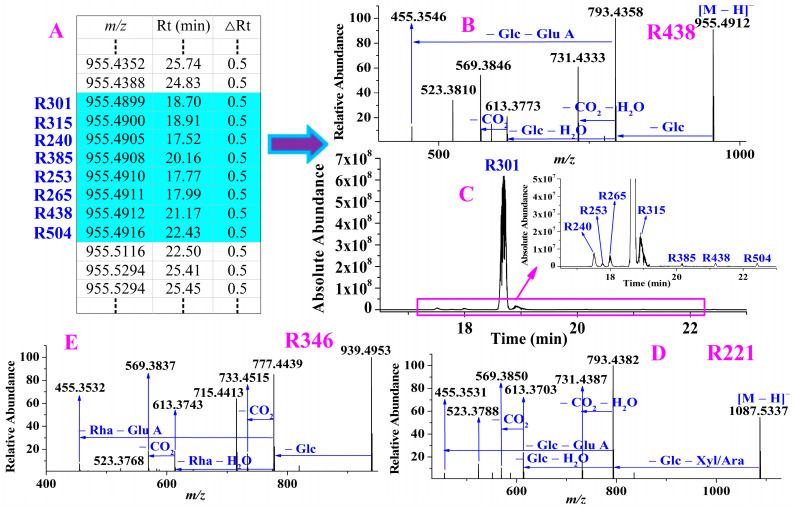
An example of characterization of ginsenosides with aglycone *m/z* 455. The precursor ions of eight compounds were reconstructed in list (**A**); performing targeted MS/MS analysis for the eight compounds such as **R438** (**B**); extraction ion chromatograms of the eight compounds (**C**); the MS/MS spectrum of potential new ginsenosides **R221** (**D**) and **R346** (**E**).

## Data Availability

The data presented in this study are available on request from the first author.

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
