# Peer review of "Comprehensive Identification of Ginsenosides in the Roots and Rhizomes of Panax ginseng Based on Their Molecular Features-Oriented Precursor Ions Selection and Targeted MS/MS Analysis"

_molecules, 2023, doi:10.3390/molecules28030941_

Round 1
Reviewer 1 Report
This study proposed a strategy based on their molecular features-oriented precursor ions selection and targeted MS/MS and gave a comprehensive identification of ginsenosides in panax ginseng. Overall, the report is fairly complete (with an up-to-date HR MS) and helpful for mass spectrometric analysis of ginsenosides. I think it's worth being published in Molecules.
Two minor issues,
1. Besides exact mass assignment (calculate the mass deviations), isotope distribution can be also helpful and used for confirmation of compound identification. Did authors check them?
2. Considering ion list illustrated inside the Figures, it is recommended that the reported m/z and LC retention time can keep with four and two decimal places respectively, just consistent with the text in article.
Author Response
Please see the attachmen
